# Associations between Disc Hemorrhage and Primary Open-Angle Glaucoma Based on Genome-Wide Association and Mendelian Randomization Analyses

**DOI:** 10.3390/biomedicines12102253

**Published:** 2024-10-03

**Authors:** Je Hyun Seo, Young Lee, Hyuk Jin Choi

**Affiliations:** 1Veterans Medical Research Institute, Veterans Health Service Medical Center, Seoul 05368, Republic of Korea; lyou7688@gmail.com; 2Department of Ophthalmology, Seoul National University College of Medicine, Seoul 03080, Republic of Korea; 3Department of Ophthalmology, Seoul National University, Hospital Healthcare System Gangnam Center, Seoul 06236, Republic of Korea

**Keywords:** primary open-angle glaucoma, Mendelian randomization, disc hemorrhage, single-nucleotide polymorphisms, glaucoma risk factor

## Abstract

**Background/Objectives:** We aimed to investigate the genetic loci related to disc hemorrhage (DH) and the relationship of causation between DH and primary open-angle glaucoma (POAG) using a genome-wide association study (GWAS) in East Asian individuals. **Methods:** The GWAS included 8488 Koreans who underwent ocular examination including fundus photography to determine the presence of DH and POAG. We performed a GWAS to identify significant single-nucleotide polymorphisms (SNPs) associated with DH and analyzed the heritability of DH and genetic correlation between DH and POAG. The identified SNPs were utilized as instrumental variables (IVs) for two-sample Mendelian randomization (MR) analysis. The POAG outcome dataset was adopted from Biobank Japan data (n = 179,351). **Results:** We found that the rs62463744 (*TMEM270*;*ELN*), rs11658281 (*CCDC42*), and rs77127203 (*PDE10A*;*LINC00473*) SNPs were associated with DH. The SNP heritability of DH was estimated to be 6.7%, with an absence of a genetic correlation with POAG. MR analysis did not reveal a causal association between DH and POAG for East Asian individuals. **Conclusions:** The novel loci underlying DH in the Korean cohort revealed SNPs in the *ELN*, *CCDC41*, and *LINC00473* genes. The absence of a causal association between DH and POAG implies that DH is a shared risk factor, rather than an independent culprit factor, and warrants further investigation.

## 1. Introduction

Glaucoma is a chronic progressive optic neuropathy that results in a typical optic nerve head (ONH) appearance and accompanying visual field (VF) loss [1]. Posterior displacement of the lamina cribrosa (LC) with blockade of axoplasmic flow are the putative features of primary open-angle glaucoma (POAG) [2,3,4,5,6,7,8]; however, its precise pathogenesis remains unknown. Elevated intraocular pressure (IOP) is a significant risk factor for POAG, demonstrated by the fact that lowering IOP is the only effective treatment for glaucoma [9]. Although glaucoma progression can occur despite a significant reduction in IOP [10,11,12], the IOP-related mechanism does not completely explain glaucoma pathogenesis [13]. Moreover, previous studies have demonstrated that various factors beyond IOP are implicated in the pathophysiology of glaucoma, disc hemorrhage (DH) being one such factor [14,15,16,17]. 

DH is a characteristic of glaucomatous optic neuropathy, which is commonly observed in low-pressure glaucoma and is uncommon in normal eyes [18,19]. Previous studies have reported DH in approximately 7.1–20% of the cases with POAG and have found it to be a precursor of glaucomatous disc changes and associated VF defects [19,20]. Additionally, a recent randomized clinical trial confirmed that DH is a risk factor for the development and progression of glaucoma [21]. Prior studies have shown that eyes with DH had greater VF deterioration than those without DH [22,23] and that recurrent DH is associated with VF deterioration [24]. Moreover, eyes with DH had a higher rate of retinal nerve fiber layer (RNFL) atrophy and conversion to first VF loss [25]. Furthermore, the Cox hazard models evaluating the risk of exposure during a particular time period support the status of DH as a risk factor for glaucoma [26]. Nonetheless, the causal association between DH and glaucoma remains ambiguous. DH is detected in early ONH changes and VF damage, and IOP-lowering medication is effective in preventing or delaying progressive VF loss. In contrast, a recent report from the Early Manifest Glaucoma Trial revealed the lack of a significant association between IOP-lowering treatment and the occurrence of DH [27]. Conversely, the identification of RNFL loss or progression prior to DH detection serves as compelling evidence of an associated finding for DH [28,29]. However, a recent study on VF showed that DH was associated with the presence and progression of central VF defect [30].

Given that the pathogenesis of DH has not been completely elucidated, identifying risk factors associated with DH may contribute to our understanding of the mechanisms involved in its occurrence. Previous studies have indicated a potential correlation between mechanical damage to the blood vessels at the LC tissue or the edge of the RNFL defect and the development of DH [31,32]. In addition, features of DH, including the retro-bulbar mechanism [33,34,35], should be investigated with respect to a causal association between DH and POAG. 

In recent years, the integration of bioinformatics and statistical methodologies in the analysis of genetic data, alongside epidemiological data, has facilitated the emergence of Mendelian randomization (MR) analysis [36]. This strategy of genetic epidemiology employs genetic variants that are linked to risk factors as instrumental variables (IVs) to investigate their causal impact on diseases and outcomes [37,38,39,40,41,42,43,44,45,46]. Thus, it is possible to investigate whether DH is a causal factor of POAG, using MR analysis. Notably, the Primary Open-Angle African American Glaucoma Genetics study did not associate the *LMX1B* gene with DH, according to the few studies that have looked into gene-related DH [47]. Consequently, we aimed to conduct a genome-wide association study (GWAS) in a Korean population cohort to identify unknown DH-related SNPs. Additionally, a two-sample MR analysis was performed to assess DH’s effect on POAG using the DH-related SNPs identified in our study as exposures, along with a GWAS summary of Biobank Japan data (n = 179,351) as the outcome dataset [48].

## 2. Materials and Methods

### 2.1. Study Design

The research protocol was approved by the Seoul National University Hospital (SNUH) Institutional Review Board (IRB No. H-1505-047-671 and IRB No. H-1804-039-935) and the Veterans Health Service Medical Center (IRB No. 2022-03-034) and conducted in accordance with the tenets of the Declaration of Helsinki. Schematic plots of the analytical study design are illustrated in Figure 1. Data were obtained from the GENIE (Gene-Environmental Interaction and Phenotype; Seoul National University Hospital Healthcare System Gangnam Center, n = 10,579) cohort [49,50,51]. A case group which included patients with DH was identified in the GENIE cohort by an ophthalmologist (HJC) using fundus photography. Since DH is transient, lasting for about 3 months, and POAG can occur independent of DH, DH detection is not always feasible. Given the high correlation between POAG and DH, it is necessary to define a control group with precision. Therefore, a control group was established wherein the absence of DH was detected using fundus photography and there was no evidence of POAG. Moreover, POAG was defined as the presence of glaucomatous optic disc changes and RNFL defect [51]. Glaucomatous optic disc changes were defined as vertical cup-to-disc ratio (C/D) > 0.7, neuroretinal rim thinning (superior or inferior rim width <0.1 times disc diameter), notching, or excavation. The baseline IOP value was defined as the mean of at least two measurements before the initiation of IOP-lowering treatment. Patients with missing genotype data or missing fundus photography images, with RNFL defects and non-glaucomatous optic disc changes (n = 152), and participants with uveitis history or diseases affecting the VF (stroke, Alzheimer’s diseases, and dementia) were excluded from this study. Participants with a diagnosis or history of any secondary glaucoma, a history of ocular trauma, a history of systemic or ocular infection, or a history of systemic or ocular use of glucocorticoids were also excluded. 

### 2.2. Genotyping

We utilized genome-wide variants genotyped with the Korea Biobank array (KoreanChip), which was developed by the Center for Genome Science at the Korea National Institutes of Health using the Affymetrix Axiom^®^ Array (Affymetrix, Santa Clara, CA, USA). Additionally, we used SHAPEIT2 v2.r904 and IMPUTE2 version 2.3.2 for haplotype phasing and imputation, respectively [52,53]. The 1000 Genomes Phase III data were used as a reference panel for imputation. Any variant with genotype call rates < 95%, minor allele frequency (MAF) values < 0.05, or in violation of the Hardy-Weinberg equilibrium (*p* < 1 × $10^{-5}$) was removed. Only SNPs with quality scores > 0.5 were retained, resulting in 3,640,889 SNPs in the GENIE cohort. The National Center for Biotechnology Information Human Genome Build 37 (hg19) was used to confirm gene locations (Figure 1).

### 2.3. Genome-Wide Association Study

The GWAS for DH was conducted using logistic regression with an additive model, PLINK 1.9. Age, sex, and ten principal component scores were included as covariates. The quantile-quantile (Q-Q) and Manhattan plots and the regional plot with LocusZoom software version 1.3 were generated for GWAS results [54].

### 2.4. Selection of the Genetic Instrumental Variables for Mendelian Randomization

SNPs associated with DH at the threshold value (*p* < 1.0 × $10^{-5}$) were utilized as IVs. SNPs were clumped using linkage disequilibrium (LD) with r^2^ < 0.001 within 10,000 kb to ensure the independence of IVs. The East Asian dataset from 1000 genomes phase III was utilized as the reference panel for computing LD for the clumping process. *F*-values were utilized to evaluate the strengths of genetic IVs using the formula *F* = *R*^2^(*n* − 2)/(1 − *R*^2^), where *n* was the sample size and *R^2^* was the proportion of variance in exposure by the genetic variants [55]. *F* values > 10 were regarded as ‘no evidence of weak instrument bias’ [56]. 

### 2.5. Statistics for Mendelian Randomization 

The MR study was based on the following assumptions for IVs: (1) they ought to reveal an essential connection in relation to exposure, (2) they must have no association with the confounds of the exposure-outcome connection, and (3) they should only impact results via exposure, indicating that there is no directional horizontal pleiotropy effect. We employed inverse-variance weighted (IVW) MR with multiplicative random effects as the main method [45,56,57]. This approach is most efficient when all genetic variants meet the three conditions for IVs [58]. However, if one or more of the variants are not valid, the estimate of the IVW analysis may be skewed [59]. Additionally, the weighted median [59], MR-Egger (with or without adjustment via the Simulation Extrapolation [SIMEX] method) regression [60,61], and the MR pleiotropy residual sum and outlier (MR-PRESSO) [62] were utilized. The weighted median method generates precise calculations of causality even when half of the instruments are erroneous [59]. The MR-Egger technique permits a suitable calculation of causal impacts regardless of a setting of pleiotropic effects, allowing for a non-zero intercept which clearly displays the average horizontal pleiotropic effects [60]. The MR-Egger with SIMEX can be utilized to rectify the bias when the presumption of no measurement error is violated [61]. The MR-PRESSO test, which detects outliers, adjusts the IVW analysis results for horizontal pleiotropy by eliminating the outliers [62]. Cochran’s Q and Rücker’s Q′ statistics were employed to evaluate the heterogeneity of IVW and MR-Egger [57,63]. Directional horizontal pleiotropy was evaluated via the MR-PRESSO global test. Hence, the results were interpreted according to the appropriate MR analysis method [64]. *p* < 0.05 for Cochran’s Q statistic, Rücker’s Q′ statistic, and MR–PRESSO global test indicated potential pleiotropy in the genetic variations. All analyses were conducted using the ‘TwoSampleMR’ and ‘Simex’ packages in R version 3.6.3 (R Core Team, Vienna, Austria).

## 3. Results

### 3.1. Characteristics of the Study Participants

Demographic data are presented in Table 1. The median age of the study population was 54.0 years in the DH group and 52.0 years in the control group, with no significant differences (*p* = 0.304, Table 1). The IOPs in the DH group were significantly higher than those in the control group (*p* < 0.05). However, the weight, body mass index, systolic blood pressure, and diastolic blood pressure were not significantly different between the DH and control groups (all *p* > 0.05). In addition, no discernible disparity was noted in the frequencies of comorbidities such as diabetes and hypertension and laboratory blood examinations such as hemoglobin A1c, fasting glucose level, and lipid level (all *p* > 0.05, Table 1).

### 3.2. Genome-Wide Association Study

The Q-Q plot showed no inflation (Figure 2A). While no SNPs reached genome-wide significance in the GWAS, SNPs with *p*-values below the threshold of 1 × 10^−5^ are listed in Table 2. SNPs such as rs62463744 (*TMEM270*;*ELN*), rs11658281(*CCDC42*), rs77127203 (*PDE10A*;*LINC00473*), and rs7589033 (*THADA*) were linked to DH (Table 2 and Figure 2). In addition, Figure 3 presents the regional association plots for the top four SNPs, showing the locations of all candidate genes within the region. The heritability of the SNPs was 6.7% with GIF = 0.993 (Table 3); however, no genetic correlation was observed between POAG and DH (*p* = 0.373). 

### 3.3. Mendelian Randomization

Nine SNPs with significance level *p* < 1.0 **×**
$10^{-5}$ were selected as DH-related IVs (Figure 4 and Appendix A). The mean *F* statistics for DH (21.73) used for MR were >10, displaying that there was a low opportunity of fragile instrument bias (Table 4).

Detailed information on the IVs used is listed in Appendix A. IVW was used as primary method because the IVs for DH were not heterogeneous in Cochran’s Q test *p* > 0.05) (Table 4). In addition, Rücker’s Q′ test from MR–Egger revealed no heterogeneity between IVs, and the MR–Egger regression intercepts showed no horizontal pleiotropic effect before (*p* > 0.05) and after SIMEX adjustment (*p* > 0.05), indicating the absence of a pleiotropic effect (Table 4). Additionally, the MR–PRESSO global test demonstrated no horizontal pleiotropy (*p* > 0.05; Table 4). DH did not show a significant causal association with POAG in all MR analyses for the East Asian population [IVW MR OR = 1.00, 95% confidence intervals (CIs): 0.99–1.01, *p* = 0.625, weighted median MR OR = 1.00, 95% CI: 0.98–1.02, *p* = 0.766, MR–Egger MR OR = 1.09, 95% CI: 0.87–1.35, *p* = 0.476, and MR–Egger (SIMEX) MR OR = 1.26, 95% CI: 0.93–1.70, *p* = 0.182, Figure 5]. 

A scatter plot displays the genetic associations of DH against genetic associations with POAG for each SNP (Figure 6).

## 4. Discussion

In the present study, we performed a GWAS to assess the effect of DH on POAG in the Korean population and identified several novel candidate loci (rs62463744 in *TMEM270*;*ELN*, rs11658281 in *CCDC42*, rs77127203 in *PDE10A*;*LINC00473*, rs7589033 in *THADA*) related to DH. The heritability of DH was estimated to 6.7%, with an absence of a genetic correlation with POAG. Additionally, the MR analysis showed that a causal association between DH and POAG was not present in the East Asian population. 

Several studies have suggested that DH is a risk factor for POAG, since it has a strong association with glaucoma. However, a “causal” risk factor with further supporting data is necessary to establish a significant association between an undocumented exposure and an outcome and is known as a correlate of the outcome or a risk marker [35]. As microinfarctions and ischemic changes can make capillaries more vulnerable to rupture [65], DH is considered a sign of ischemic optic nerve damage. In addition, systemic hypertension or hypotension, diabetes, migraine, or medication, including platelet aggregation inhibitors, have been suggested to be associated with DH [66]. Stretching of the vessels from posterior migration of the LC [2] and vessel damage due to mechanical collapse have also been suggested as possible causes. Additionally, the hypotheses that 1) the presence of DH at the affected LC is not consistently observed in nearby regions [67] and 2) the detection of DH occurs after to the advancement of glaucoma [35] raise concerns regarding the potential causal relationship between DH and glaucoma. Therefore, further investigation is necessary to address these hypotheses.

GWAS is a powerful tool that allows us to scan the genome comprehensively to identify genetic variants associated with traits or diseases [68]. Thus, we identified specific genetic markers that contribute to the risk of DH, providing a genetic foundation which not only enhances our understanding of the biological pathways involved but also helps in identifying potential targets for therapeutic intervention. MR uses genetic variants as IVs to estimate the causal effect of an exposure (DH) on an outcome (POAG) while minimizing confounding and bias, which are typical in observational studies. Consequently, MR allows us to infer causality by mimicking the conditions of a randomized controlled trial by leveraging the genetic variants associated with DH identified from our GWAS as tools. This approach is particularly useful in understanding whether the pathways leading to DH contribute causally to the development of POAG, rather than simply being associated with it. To date, several GWAS studies have identified various genetic loci and risk factors that are associated with glaucoma [69,70]. DH is often observed in the ONH in patients with POAG and is considered an important clinical sign of glaucoma progression. While GWAS studies have explored the genetics of glaucoma, limited studies have specifically evaluated the genetics of DH in patients with glaucoma. This may be due to the challenge of obtaining large enough sample sizes of patients with DH, as it is a relatively rare event in patients with glaucoma, as well as a limited phenotype, since a large genetic cohort does not contain phenotypes such as DH. Notably, the GWAS for DH highlighted the involvement of rs62463744 (*TMEM270*;*ELN*), rs11658281 (*CCDC42*), rs77127203 (*PDE10A*;*LINC00473*), and rs7589033 (*THADA*) as SNPs linked to DH. The identification of these genetic markers underscores the multifactorial nature of DH and suggests a genetic predisposition that may contribute to its pathogenesis.

The rs62463744 SNP is present in the *TMEM270* and *ELN* (Elastin) genes. The *ELN* gene encodes a protein elastin fiber which is present in the extracellular matrix and provides elasticity to tissues, including the heart, skin, and blood vessels [71]. Additionally, it provides recoil tissue for vascular elasticity [72]. Consequently, genetic mutations in the *ELN* gene that reduce elastin protein levels are associated with focal arterial stenosis or narrowing of the arterial lumen. Although the genetic analysis of DH is limited, the presence of a specific variant of the *ELN* gene has been found to indicate a heightened vulnerability among to the development of intracranial aneurysms in individuals of East Asian descent [73,74]. According to previous studies showing abnormal elastin synthesis in an experimental model glaucomatous optic neuropathy in monkeys, this variant is specific to elevated IOP and not secondary to axonal loss [75]. These abnormalities in elastin may be related to the development and progression of glaucoma in patients with DH. However, as one previous study reported a lack of association of polymorphisms in elastin with pseudo exfoliation syndrome and glaucoma [76], these results suggest that *ELN* may be associated with DH as vascular factor. In addition, the *TMEM270* gene (Transmembrane Protein 270) is related to hemorrhage and aortic measurement. The rs11658281 SNP is present in the *CCDC42* (Coiled-coil domain-containing protein 42) gene and is related to sex hormones, type 2 diabetes, and vision disorders. The rs77127203 SNP is related to *PDE10A* (Phosphodiesterase 10A) and *LINC00473* (Long Intergenic Non-Protein Coding RNA 473) genes. *PDE10A* plays a role in signal transduction by regulating the intracellular concentration of cyclic nucleotides, which is associated with platelet count, blood pressure, thyroid function, refractive errors, and cataract. In addition, *the LINC00473* gene, affiliated with the lncRNA class, is related to some phenotypes, such as mean arterial pressure, systolic blood pressure, and platelet count. The rs7589033 SNP is in the *THADA* (*THADA Armadillo Repeat Containing*) gene and encodes a protein which is likely involved in the death receptor pathway and apoptosis. The *THADA* gene is related to phenotypes such as platelet count and type 2 diabetes. According to a previous study [77], the *THADA* gene is related with IOP in glaucoma GWASs.

These (*TMEM270*, *ELN*, *CCDC42*, *PDE10A*, and *LINC00473*) genes associated with DH are involved in various cellular processes, including vascular regulation, extracellular matrix maintenance, and intracellular signaling pathways. Therefore, elucidating how variations in these genes contribute to the vulnerability of the optic nerve head and retinal vasculature could enhance our understanding of the mechanistic links between DH and glaucoma. Moreover, the genetic analysis of DH and glaucoma has the potential to inform personalized medicine or targeted intervention. Vascular management intervention in addition to IOP-lowering treatment may be considered when this customized treatment has more evidence and a higher propensity to be associated with vascular factors. Thus, identifying individuals with a genetic predisposition to DH and glaucoma may enable earlier intervention and more targeted treatment strategies. Additionally, it may facilitate the development of novel therapeutic interventions aimed at mitigating the genetic risk factors associated with these conditions.

One of the primary features of our work is the comprehensive characterization of a DH phenotype, which was achieved through the analysis of phenotypes acquired from a standardized fundus examination undertaken at a single institution by an ophthalmologist. Unvalidated participants were not included as the control group, and the phenotype was defined strictly. Furthermore, given that Koreans are classified as a homogeneous ethnic group and that the MR analysis was conducted using Japanese data, it is anticipated that the impact of racial disparities would be minimal for East Asian individuals. Nevertheless, a notable constraint lies in the limited scope of the replication cohort due to the absence of comparable investigations. Additionally, our study exclusively involved East Asian individuals; therefore, the genetic variants identified may have different allele frequencies or effects in other populations due to genetic diversity and environmental factors. Although this focus allows for a clearer understanding of genetic predispositions within this demographic, it indeed limits the immediate applicability of our findings to other ethnic groups. Such population-specific results might not capture genetic variants influential in other ethnicities, potentially missing broader genetic insights related to DH and POAG. Therefore, we suggest that further studies be conducted in diverse populations to either confirm or refine our findings. Additionally, cross-population studies could be employed to explore the genetic architecture across different ethnic groups, enhancing the understanding of how these findings can be applied globally. Furthermore, it is important to note that the proof of causality is constrained by the restricted availability of significant IVs. However, given the absence of prior GWAS research on DH and causation verification of DH and POAG, the anticipated outcome of this study on DH will contribute to the fundamental understanding of the risk variables associated with POAG, albeit with certain limitations.

## 5. Conclusions

We identified candidate novel genetic loci that were related to DH. This genetic analysis of DH and glaucoma provides valuable insights into the complex interplay of genetic factors in the pathogenesis of these ocular disorders. In addition, the non-significant causal association between DH and POAG implies that the role of DH as an indicator of glaucoma-related phenomenon rather than a risk factor for glaucoma, supported by researchers and increasing evidence. Several studies have reported that DH is a risk factor for glaucoma and glaucoma progression [21,78]; however, the results of our study suggest that DH is a biomarker for glaucoma as a shared risk factor rather than an independent culprit factor. However, further research is warranted to validate these genetic associations, unravel the molecular mechanisms involved, and translate these findings into clinical applications for more effective management and personalized care in individuals at risk for DH and glaucoma. In addition, covariate analysis of factors related to DH is necessary to determine whether it is a fundamental risk factor for glaucoma.

## Figures and Tables

**Figure 1 biomedicines-12-02253-f001:**
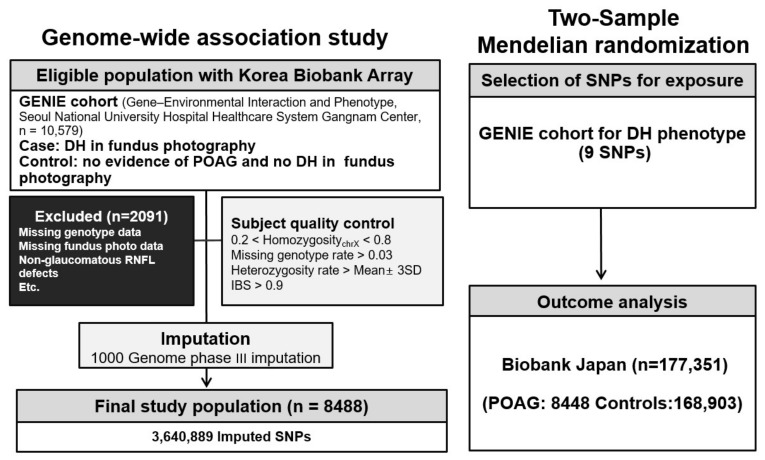
Diagram presentation of the study design. POAG, primary open-angle glaucoma; DH, disc hemorrhage; GENIE cohort, Gene-Environmental Interaction and Phenotype; SNP, single-nucleotide polymorphism.

**Figure 2 biomedicines-12-02253-f002:**
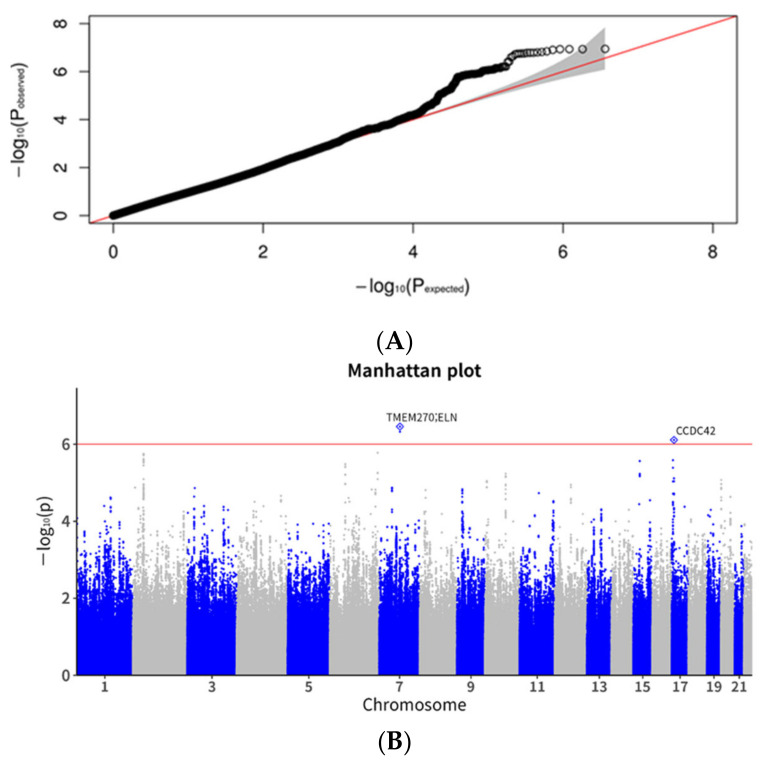
Quantile-quantile and Manhattan plots for disc hemorrhage in the genome-wide association study. (**A**). Quantile-quantile (Q-Q) plot. The expected line is shown in red and confidence bands are shown in gray. (**B**). Manhattan plot. The red line indicates the preset threshold of *p* = 1.0 × 10^−6^.

**Figure 3 biomedicines-12-02253-f003:**
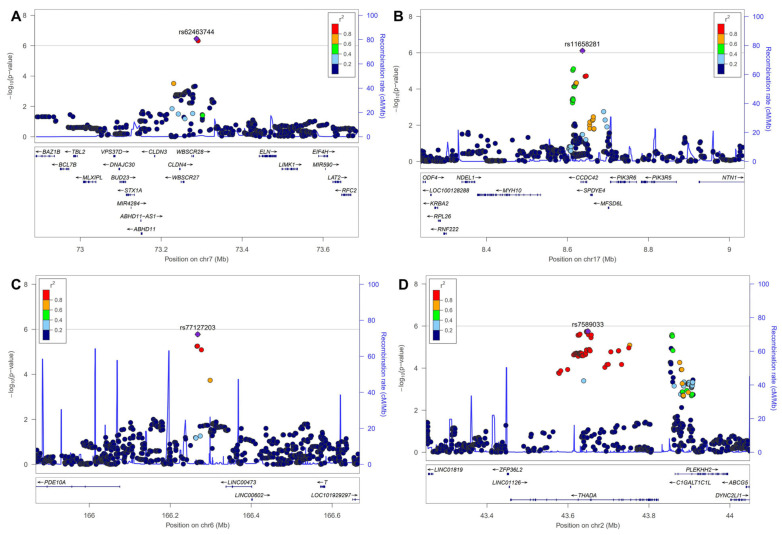
Regional association plots for top 4 SNPs. SNP, single-nucleotide polymorphism. (**A**): rs62463744, (**B**): rs11658281, (**C**): rs77127203, (**D**): rs7589033.

**Figure 4 biomedicines-12-02253-f004:**
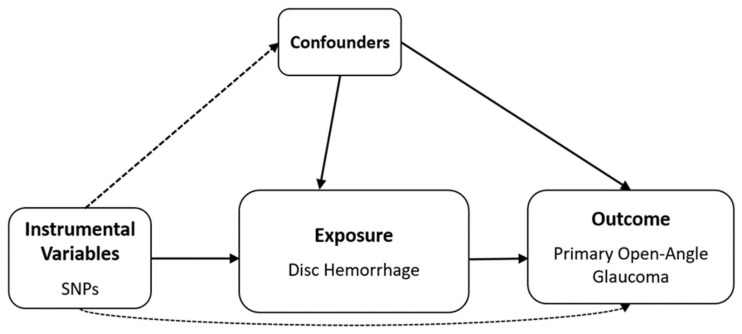
Schematic design of Mendelian randomization analysis. SNP, single-nucleotide polymorphism. Solid lines indicate the presence of an association, dashed lines indicate the absence of an association.

**Figure 5 biomedicines-12-02253-f005:**
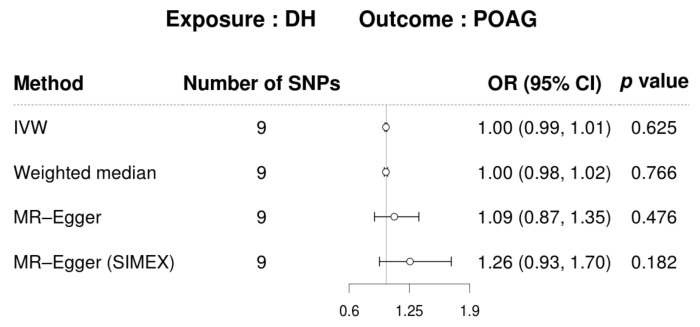
MR visualizations of the effect of DH on POAG. CI, confidence interval; DH, disc hemorrhage; POAG, primary open-angle glaucoma; OR, odds ratio; SIMEX, simulation extrapolation; SNP, single-nucleotide polymorphism.

**Figure 6 biomedicines-12-02253-f006:**
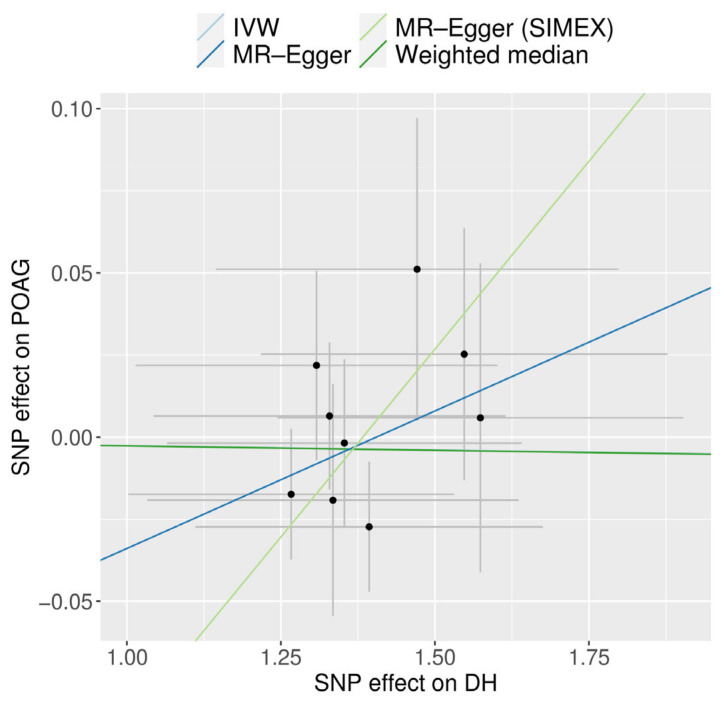
Scatter plot of MR analyses using different MR methods evaluating the impact of the existence of DH on POAG. Light blue, dark blue, light green, and dark green regression lines represent IVW, MR-Egger, MR-Egger (SIMEX), and weighted median estimate, respectively. SNP, single-nucleotide polymorphism; DH, disc hemorrhage; POAG, primary open-angle glaucoma; MR, Mendelian randomization; SIMEX, simulation extrapolation.

**Table 1 biomedicines-12-02253-t001:** Baseline characteristics of enrolled patients.

	Disc Hemorrhage N = 31	Control N = 8457	*p*-Value *
Age (years)	54.0 (46.0, 58.5)	52.0 (46.0, 58.0)	0.304
Sex			0.414
Male n (%)	21 (67.7%)	4981 (58.9%)	
Female n (%)	10 (32.3%)	3476 (41.1%)	
IOP (mmHg)			
Right eye	13 (13, 15)	12 (10, 14)	0.023
Left eye	15 (13, 16)	13 (11, 15)	0.003
Mean IOP	14.5 (12.0, 15.5)	12.5 (10.5, 14.5)	0.007
Higher IOP of both eyes	15 (13, 16)	13 (11, 15)	0.004
Weight (kg)	63.7 (56.4, 72.8)	64.8 (55.6, 73.0)	0.698
BMI (kg/m)	23.58 (21.10, 25.06)	23.18 (21.14, 25.13)	0.783
Systolic blood pressure (mmHg)	120.0 (103.5, 128.5)	115 (106, 125)	0.535
Diastolic blood pressure (mmHg)	81.0 (70.0, 87.5)	76 (69, 83)	0.062
Comorbidity			
Diabetes, n (%)	1 (3.23%)	296 (3.51%)	1
Hypertension, n (%)	2 (6.45%)	982 (11.6%)	0.573
Laboratory examination			
HbA1c (%)	5.50 (5.35, 5.75)	5.6 (5.4, 5.8)	0.536
Fasting glucose (mg/dL)	95 (91, 100)	96 (90, 103)	0.742
Total cholesterol (mg/dL)	183.0 (169.0, 202.5)	193 (171, 216)	0.286
LDL cholesterol (mg/dL)	112 (100, 142)	121 (101, 142)	0.517
HDL cholesterol (mg/dL)	52.0 (46.5, 61.0)	52 (45, 60)	0.72
Triglyceride (mg/dL)	79 (61, 112)	90 (63, 133)	0.196

* Continuous variables were summarized using medians [interquartile ranges] and compared using the Mann-Whitney U test, while categorical variables were summarized as n (%) and analyzed using the chi-square test and Fisher’s exact test. IOP, intraocular pressure; BMI, body mass index; HbA1c, glycosylated hemoglobin; LDL, low-density lipoprotein; HDL, high-density lipoprotein.

**Table 2 biomedicines-12-02253-t002:** SNPs reaching suggestive significance at *p* < 1 × 10^−5^ in GWAS.

Chr	SNP	Position	Allele	MAF	OR	L95	U95	*p*	Mapped Genes
7	rs62463744	73287385	C/T	0.06888	4.87	2.65	8.95	3.49 × 10^−7^	*TMEM270*;*ELN*
17	rs11658281	8637070	T/G	0.1353	4.03	2.32	7.00	7.78 × 10^−7^	*CCDC42*
6	rs77127203	166267889	A/G	0.2755	3.55	2.11	5.96	1.67 × 10^−6^	*PDE10A*;*LINC00473*
2	rs7589033	43649228	T/C	0.05157	4.83	2.53	9.20	1.77 × 10^−6^	*THADA*
2	rs58526585	43857040	T/C	0.05278	4.76	2.48	9.11	2.59 × 10^−6^	*THADA*;*PLEKHH2*
17	rs113460962	4850748	C/T	0.09861	3.87	2.20	6.80	2.60 × 10^−6^	*PFN1*
15	rs76143071	50723310	C/T	0.0526	4.70	2.46	8.98	2.74 × 10^−6^	*USP8*
6	rs9462784	42202149	C/T	0.1106	3.78	2.16	6.61	3.31 × 10^−6^	*TRERF1*
17	rs2302320	4796656	C/T	0.1821	3.37	2.01	5.64	4.10 × 10^−6^	*MINK1*
10	rs140873075	80392732	G/C	0.05223	4.42	2.33	8.41	5.84 × 10^−6^	*LINC00595*;*ZMIZ1-AS1*
15	rs3803373	50792874	A/G	0.07226	4.09	2.22	7.55	6.40 × 10^−6^	*USP50*;*USP8*
20	rs78583358	1840763	T/G	0.09542	3.70	2.08	6.58	8.50 × 10^−6^	*LOC100289473*;*SIRPA*
10	rs78617863	5523330	G/C	0.08089	3.82	2.11	6.91	9.10 × 10^−6^	*NET1*;*CALML5*
17	rs112718881	4855824	T/C	0.05742	4.43	2.29	8.56	9.32 × 10^−6^	*ENO3*

Chr, chromosome; OR, odds ratio; L95, lower limit of 95% confidence interval; U95, upper limit of 95% confidence interval; MAF, minor allele frequency; SNP, single-nucleotide polymorphisms; GWAS, genome-wide association study.

**Table 3 biomedicines-12-02253-t003:** Estimated heritability and genetic correlation.

Trait	Heritability		Genetic Correlation with POAG	
	*h*^2^ ± SE	GIF	GC ± SE	*p*
DH	0.067 ± 0.056	0.993	0.257 ± 0.289	0.373

DH, disc hemorrhage; SE, standard error; GIF, genomic inflation factor; GC, genetic correlation; *p*, *p*-value from LDSC (linkage disequilibrium score) regression; POAG, primary open-angle glaucoma.

**Table 4 biomedicines-12-02253-t004:** Heterogeneity and horizontal pleiotropy of the instrumental variables.

			Heterogeneity		Horizontal Pleiotropy				
						MR-Egger		MR-Egger (SIMEX)	
N	F	I^2^ (%)	*p*-Value *	*p*-Value #	*p*-Value ^†^	Intercept, β(SE)	*p*-Value	Intercept, β(SE)	*p*-Value
9	21.73	39.56	0.743	0.717	0.728	−0.118 (0.151)	0.462	−0.316 (0.21)	0.175

*: Cochran’s Q test from IVW, #: Rücker’s Q’ test from MR-Egger, †: MR-PRESSO global test. N, number of instruments; F, mean F statistic; IVW, inverse-variance weighted; MR, Mendelian randomization; PRESSO, pleiotropy residual sum and outlier; SIMEX, simulation extrapolation; β, beta coefficient; SE, standard error.

## Data Availability

The datasets used and/or analyzed in the current study are available from Biobank Japan (BBJ; https://pheweb.jp/) [43]. The datasets generated and analyzed during the current study are available from the corresponding author upon reasonable request.

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
