# Peer review of "Associations between Disc Hemorrhage and Primary Open-Angle Glaucoma Based on Genome-Wide Association and Mendelian Randomization Analyses"

_biomedicines, 2024, doi:10.3390/biomedicines12102253_

Round 1

Reviewer 1 Report

Comments and Suggestions for Authors

To explore the genetic loci related to disc hemorrhage (DH) and the causal association between DH and primary open-angle glaucoma (POAG), the authors used a genome-wide association study (GWAS) in East Asian individuals. They reported that they didn’t find any relationship between DH and POAG. However, they found that the top four SNPs may relate to DH, which is noteworthy, although they didn’t investigate the details at the molecular level. They used enough data to conclude. In sum, I recommend publishing this manuscript in the biomedicines.   

Author Response

Reviewer 1

To explore the genetic loci related to disc hemorrhage (DH) and the causal association between DH and primary open-angle glaucoma (POAG), the authors used a genome-wide association study (GWAS) in East Asian individuals. They reported that they didn’t find any relationship between DH and POAG. However, they found that the top four SNPs may relate to DH, which is noteworthy, although they didn’t investigate the details at the molecular level. They used enough data to conclude. In sum, I recommend publishing this manuscript in the biomedicines.

Response 1: Thank you for your valuable comments. 

Reviewer 2 Report

Comments and Suggestions for Authors This manuscript uses genome-wide association and Mendelian randomisation analyses on a Korean cohort to investigate the relationship between disc hemorrhage (DH) and primary open-angle glaucoma (POAG). It aims to identify genetic loci related to DH and study the underlying relationship between DH and POAG. The following are comments to improve the overall presentation of the manuscript. 1.     The introduction needs more elaboration on the clinical and genetic significance of disc haemorrhage in glaucoma by adding more recent references on genetic correlations between DH and vascular conditions. 2.     Discuss the introduction and explain why the genetic factors you are particularly relevant for understanding glaucoma progression and DH. 3.     Provide more details on the inclusion and exclusion criteria for selecting participants in the case and control groups. 4.     Explain the rationale behind using GWAS and the Mendelian Randomization model. 5.     Discuss the clinical relevance of the identified SNPs, how they contribute to understanding glaucoma progression, and how genetic variations could influence disease susceptibility and treatment strategies. 6.     The discussion section needs further discussion on the genetic findings and their potential for application in personalised medicine or targeted interventions. 7.  Discuss the possible limitations of the study, such as population-specific findings, and how they could influence the generalizability of your results. 8.     In the conclusion, put more emphasis on the significance of the results in advancing the understanding of DH and glaucoma and suggest possible directions for future research. Comments on the Quality of English Language

Moderate editing is required to improve sentence structure and flow. 

Author Response

Reviewer 2

This manuscript uses genome-wide association and Mendelian randomisation analyses on a Korean cohort to investigate the relationship between disc hemorrhage (DH) and primary open-angle glaucoma (POAG). It aims to identify genetic loci related to DH and study the underlying relationship between DH and POAG. The following are comments to improve the overall presentation of the manuscript.

  • Our change in written in red text. In addition, an English editing service (Editage services:www.editage.co.kr) reviewed the manuscript for typo.

Comment 1: The introduction needs more elaboration on the clinical and genetic significance of disc haemorrhage in glaucoma by adding more recent references on genetic correlations between DH and vascular conditions.

Response 1: Thank you for your pertinent comments. As suggested, we have elaborated on the clinical and genetic significance of disc haemorrhage in glaucoma (lines 46–52 and 77–79).

“Previous studies have reported DH in approximately 7.1% – 20% of the cases with POAG, which was a precursor of glaucomatous disc changes and associated VF defects [19,20]. Additionally, a recent randomized clinical trial has confirmed that DH is a risk factor for the development and progression of glaucoma [21]. Prior studies have shown that eyes with DH had greater VF deterioration than those without DH [22,23] and that recurrent DH is associated with VF deterioration [24].”

“Notably, the Primary Open-Angle African American Glaucoma Genetics study did not associate LMX1B gene with DH, according to the few studies that have looked into gene-related DH [47].”

Comment 2.     Discuss the introduction and explain why the genetic factors you are particularly relevant for understanding glaucoma progression and DH.

Response 2: Thank you for pertinent comments. As per your suggestion, we have included an explanation regarding this aspect in the discussion section (lines 306–312).

“According to previous studies showing abnormal elastin synthesis in an experimental model glaucomatous optic neuropathy in monkeys, is specific to elevated IOP and not secondary to axonal loss [75]. These abnormalities in elastin may be related to the development and glaucoma progression in patients with DH. However, as one previous study reported the lack of association of polymorphisms in elastin with pseudo exfoliation syndrome and glaucoma [76], these results suggest that ELN may be associated with DH as vascular factor.”

Comment 3.     Provide more details on the inclusion and exclusion criteria for selecting participants in the case and control groups.

Response 3: We added more details on the inclusion and exclusion criteria (lines 104–110) and updated figure 1 for better understanding.

“Patients with missing genotype data or missing fundus photography images, with RNFL defects and non-glaucomatous optic disc changes (n=152) and participants with uveitis history, diseases affecting the VF (stroke, Alzheimer’s diseases, and dementia) were excluded from the study. Participants with a diagnosis or history of any secondary glaucoma, a history of ocular trauma, a history of systemic or ocular infection or a history of systemic or ocular use of glucocorticoids were also excluded.”

Comment 4.     Explain the rationale behind using GWAS and the Mendelian Randomization model.

Response 4: Thank you for your comment requesting clarification on the rationale behind employing Genome-Wide Association Studies (GWAS) and Mendelian Randomization (MR) in our study. We have included an explanation regarding this selection in the Discussion section (lines 275–286).

“GWAS is a powerful tool that allows us to scan the genome comprehensively to identify genetic variants associated with traits or diseases [68]. Thus, we identified specific genetic markers that contribute to the risk of DH, providing a genetic foundation which not only enhances our understanding of the biological pathways involved but also helps in identifying potential targets for therapeutic intervention. MR uses genetic variants as IVs to estimate the causal effect of an exposure (DH) on an outcome (POAG) while minimizing confounding and bias typical in observational studies. Consequently, MR allows us to infer causality by mimicking the conditions of a randomized controlled trial by leveraging the genetic variants associated with DH identified from our GWAS as tools. This approach is particularly useful in understanding whether the pathways leading to DH contribute causally to the development of POAG, rather than simply being associated with it.”

Comment 5.     Discuss the clinical relevance of the identified SNPs, how they contribute to understanding glaucoma progression, and how genetic variations could influence disease susceptibility and treatment strategies.

Response 5: Thank you for your pertinent comment. We have added an explanation regarding the clinical relevance of the identified SNPs in the discussion (line 306–312 and 324–325)

“According to previous studies showing abnormal elastin synthesis in an experimental model glaucomatous optic neuropathy in monkeys, is specific to elevated IOP and not secondary to axonal loss [75]. These abnormalities in elastin may be related to the development and glaucoma progression in patients with DH. However, as one previous study reported the lack of association of polymorphisms in elastin with pseudo exfoliation syndrome and glaucoma [76], these results suggest that ELN may be associated with DH as vascular factor.”

“According to a previous study [77], the THADA gene is related with IOP in glaucoma GWAS.”

Comment 6.     The discussion section needs further discussion on the genetic findings and their potential for application in personalised medicine or targeted interventions.

Response 6: Thank you for your pertinent comment and valuable suggestion. We included a discussion regarding the potential of our findings for application in personalised medicine or targeted interventions in discussion (line 331–334).

“Moreover, the genetic analysis of DH and glaucoma has the potential to inform personalised medicine or targeted intervention. Vascular management intervention in addition to IOP-lowering treatment may be considered when this customized treatment has more evidence and a higher propensity to be associated with vascular factors.”

Comment 7.  Discuss the possible limitations of the study, such as population-specific findings, and how they could influence the generalizability of your results.

Response 7: Thank you for your insightful comment regarding the need for potential limitations stemming from the focus on an East Asian population in our study. We elaborated on this in the discussion section (line 347–358).

“Additionally, our study exclusively involved East Asian individuals; therefore, the genetic variants identified may have different allele frequencies or effects in other populations due to genetic diversity and environmental factors. Although this focus allows for a clearer understanding of genetic predispositions within this demographic, it indeed limits the immediate applicability of our findings to other ethnic groups. Such population-specific results might not capture genetic variants influential in other ethnicities, potentially missing broader genetic insights related to DH and POAG. Therefore, we suggest that further studies be conducted in diverse populations to either confirm or refine our findings. Additionally, cross-population studies could be employed to explore the genetic architecture across different ethnic groups, enhancing the understanding of how these findings can be applied globally.”

Comment 8.     In the conclusion, put more emphasis on the significance of the results in advancing the understanding of DH and glaucoma and suggest possible directions for future research.

Response 8: Thank you for your pertinent comment. We have emphasized the significance if our results in the Conclusion section as per your suggestion (line 366–377).

“In addition, the non-significant causal association between DH and POAG implies that the role of DH as an indicator of glaucoma-related phenomenon rather than a risk factor for glaucoma, supported by researchers and increasing evidence. Several studies have reported that DH is a risk factor for glaucoma and glaucoma progression [21,78], however the results of our study suggest that DH is a biomarker for glaucoma, as a shared risk factor, rather than an independent culprit factor. However, further research is warranted to validate these genetic associations, unravel the molecular mechanisms involved, and translate these findings into clinical applications for more effective management and personalized care in individuals at risk for DH and glaucoma. In addition, covariate analysis of factors related to DH is necessary to determine whether it is a fundamental risk factor for glaucoma.”

Round 2

Reviewer 2 Report

Comments and Suggestions for Authors

The authors addressed the raised comments appropriately.